# A Holistic Vision of the Academic and Sports Development of Elite Spanish Track and Field Athletes

**DOI:** 10.3390/ijerph20065153

**Published:** 2023-03-15

**Authors:** Eva Asensio Castañeda, Rafael M. Navarro, José L. Chamorro, Jonathan Ospina-Betancurt

**Affiliations:** 1Faculty of Sports Sciences, Universidad Europea de Madrid, 28670 Madrid, Spain; 2Hum-878 Research Team, Health Research Centre, Department of Psychology, University of Almería, 04120 Almería, Spain; 3Valoración del Rendimiento Deportivo, Actividad Física y Salud, y Lesiones Deportivas (REDAFLED), Universidad de Valladolid, 42005 Soria, Spain

**Keywords:** athletic career development, career transition, dual career, high performance, sport

## Abstract

Background: Combining sports and studies could be stressful, but beneficial for the athletes’ career development. This study explores resources and barriers in the combination of the sports and academic life spheres during the lifespan of elite Spanish track-and-field athletes. Methods: Seven elite Spanish track-and-field athletes participated in a semistructured interview about their experiences to establish a dual career combining sports and studies/work. Interpretive phenomenological analysis (IPA) followed to analyze data. Results: Findings show that elite Spanish track-and-field athletes face educational and institutional barriers to developing a dual career. Time management, social support, and additional resources could play a key role in the success or failure to develop a dual career. Conclusions: This study shows that, to overcome dual-career barriers, athletes are resourceful if social support is provided at both the micro (coaches, families, etc.) and macro (political and educational institutions) levels. It also shows that pursuing an academic career can help in alleviating inherent tensions to athletic life and in finding personal balance.

## 1. Introduction

Being a high-performance athlete is a demanding and challenging task for most children who start in sports. Even among successful young athletes, numerous studies have indicated that only a small portion of them become professional athletes. Athletes often view the transition from junior to senior as the most difficult part of their athletic career, and many admit to failing [1]. For example, data show that, in one of the most popular sports in Europe, football/soccer, fewer than 10% of elite youth football players will ever play for a professional team [2,3]. There are similar figures in less popular and less widely participated sports. In a study of 167 national champion Belgian track-and-field athletes between the ages of 14 and 18, only 17% were competing at an elite level five years later [4].

These data suggest the need to develop alternative career paths or spheres of life beyond athletic careers. One proposal that has gained attention in recent decades among researchers, sports professionals, and even political and sports institutions is dual careers (DCs) [5,6,7,8]. A DC can briefly be defined as “a career focused on sport and education or work” [9]. Dual careers arise from the need to develop athletic career models with a holistic perspective of the athlete, taking into account the challenges, resources, barriers, conflicts, and facilitators of other levels of personal development beyond sports, such as the psychological, social, academic, and vocational levels. This broader exploration of the “whole career/whole person” provides a richer understanding of the development of an athlete’s career [10].

One of the models that arose in response to the need to holistically explore athletic careers is the developmental model of transitions faced by athletes [11], which was later expanded and updated to become the holistic athletic career (HAC) model [12]. This model represents the development of an athletic career along with the development of five more levels within the same person; each of the levels has consecutive stages that interact with each other. For example, the athletic level is composed of the stages and transitions within the athlete’s athletic development: an initiation stage, a development stage during which young talented athletes intensify their training and competitions, the perfection or mastery stage that coincides with high-performance athletic environments, and the stage of the retirement from the sport. However, HAC also proposes five levels of personal development that interact with athletic development: the psychological, psychosocial, academic/vocational, financial, and legal levels. For example, within the psychological level, the characteristics of adolescence aid in having a broader understanding of the athletic experiences that occur at this age. Likewise, the psychosocial level highlights the importance of the social environment in athletic development, and the academic/vocational level reflects the interaction of athletic development with other identities within the same person.

That HAC proposes an academic development level within the study of athletic careers gives us the opportunity to frame athletic and academic career development within the career development of athletes. In fact, many studies were carried out on the compatibility of the athletic and academic spheres in athletes [8,13]. The scientific literature highlights that a significant number of athletes develop an athletic career that, combining sports and studies, entails a series of demands at different levels of the HAC model, and athletes have a series of internal and external resources and barriers that mark the success or failure of the development of the athletic career [14]. According to Stambulova [15], effective coping during career development depends on the balance between resources (factors that facilitate the coping process) and barriers (factors that interfere with effective coping) that athletes have available. For this reason, it is essential to explore resources to maximize them and make athletes more resourceful to cope with dual-career challenges. For example, various studies on other sports showed that athletes rely on personal (i.e., self-discipline, autonomous motivation) and external (i.e., academic flexibility, academic support, sporting support systems) resources to cope with the difficulty of developing a dual career [16,17,18]. The literature shows that the main barriers to balancing the sport and academic life spheres are the lack of time, financial support, inflexible schedules, and fatigue [19].

However, cultural differences and the diversity of existing sports demand studies that are framed in specific cultures and sports [8,20]. Recently, Mejías et al. [21] published a taxonomy of dual-career development environments (DCDEs) in Spain. The results identified three macrosystems in which DCDEs are organized: (a) public sports centers that are characterized by having institutes dedicated exclusively to the student athletes within their facilities, (b) private sports clubs encompassing professional and amateur specialized and multisport high-performance clubs with their models of CD counseling, and (c) educational centers including universities that provide tutoring services, and flexibility in exam and class schedules. Generally, elite Spanish track-and-field athletes train in public sports (i.e., high-performance) centers, but also belong to private sports clubs. In this sense, studies that explore the experiences of track-and-field athletes who are developing their athletic career are scarce. The characteristics of this sport and the studies carried out on career development in track-and-field athletes [22,23,24] highlight that they may have demands, resources, and barriers to developing an athletic career that are different from those of other athletes. To the best of our knowledge, no studies have explored the experiences of athletic career development in elite track-and-field athletes in the Spanish context. For these reasons, the aim of this study is to explore resources and barriers in the combination of the athletic and academic life spheres during the lifespan of elite Spanish track-and-field athletes.

## 2. Materials and Methods

### 2.1. Participants

A convenience sample was used. A total of seven elite athletes were interviewed comprising two women and five men. The main inclusion criteria were that the athletes had a job or studied while training and competing, were recognized by the Superior Sports Council as high-level athletes, and were over 18 years old. Of the total number of participants, two women were undergraduate university students; of the men, two were master’s university students, two were in their second year of high school (before university), and one worked in the private sector (university studies completed) (see Table 1).

### 2.2. Instrument and Data Collection

The used instrument was the semistructured interview script according to Sparkes and Smith (2014), providing greater flexibility in the collection of information, so that we could reach saturation in the initially proposed dimensions. The interview script was adapted from other studies published by the research group [25,26]. In the development of the interview, we attempted to cover aspects of the three factors proposed by De Bosscher et al. [27]: micro (physical and psychological characteristics, family, friends, technical team, etc.), macro (social and cultural context, geographical aspects, etc.), and meso (existing sports policies), and aspects of the six levels of the holistic sports career model proposed by Wylleman [12]: sport, psychological, psychosocial, academic, economic, and legal. Following the aforementioned factors and levels, the interviewees were asked about general aspects of their sports careers. The second part collected information on the changes that had occurred during the transition to the elite level and the skills or tools used to face these changes. The athletes were also asked about the support and barriers in their sports careers. Lastly, there are considerations that, according to personal experience, should be taken into account in the dual career that were not addressed in the interview. All interviews had an approximate duration of 45 min, and were recorded and later transcribed.

### 2.3. Procedure

The project procedure was approved by the ethics committee of the European University of Madrid (code: CIPI/19/082). Before starting the interviews, the team consulted with coaches from a high-performance sports center, chosen to verify the feasibility of conducting interviews with their athletes, and we reviewed the database of top-level athletes from the Faculty of Physical Activity, Sports, and Physiotherapy at the European University of Madrid. The research team had contacted the athletes, explained the project’s objectives, the structure of the interview, and the approximate time, and provided them with the option to withdraw from the interview at any time. The seven athletes who had agreed to participate signed an informed consent form, and all possible questions from the investigators were resolved. Lastly, all interviews were conducted by a single investigator in an appropriate room.

### 2.4. Data Analysis

For the analysis and treatment of the data collected in the interviews, computer-assisted qualitative data analysis software ATLAS.ti 9 for Windows was used. With this qualitative analysis software, a hermeneutic unit was created. Interpretive phenomenological analysis (IPA) followed whose methodology aims to understand the experience of each participant in the research, and recommends working with small samples [28], which was in line with the study’s objective. The data were analyzed following the strategy and steps suggested by Smith et al. [29] for IPA, which was used in other sport studies [30,31]. The first step involved multiple readings of the initial interview. Then, the preliminary analysis of the semantic content and language was conducted to identify the initial notes. Next, emerging themes were identified, and connections were established between them. This process was repeated for each subsequent interview, and patterns were identified across cases. Lastly, all raw textual codes were examined to create subordinate and superordinate themes across all cases. Six dimensions emerged following the three factors proposed by De Bosscher et al. [27] on the factors that could impact the possibility of achieving sports success, and the five levels of the holistic sports career model proposed by Wylleman [12]. All interviews were coded by an investigator using the proposed dimensions. Subsequently, they were coded by two additional investigators. Triangulation yielded coding in four dimensions with respective categories (see Table 2) that captured the most relevant aspects in the dual career of Spanish athletes. Lastly, internal coherence analysis was performed to increase the accuracy and corroborate the consistency [32].

## 3. Results

The results are presented on the basis of the four dimensions (see Table 2) and the respective categories that were the most relevant in the dual careers of Spanish athletes: supports, barriers, time management, and resources.

### 3.1. Supports

#### 3.1.1. Family

The seven participants almost unanimously expressed support from their families, although one of the athletes had only partial support from the family, and were conditioned by the family environment in which they had grown up and lived.

“... *my mother has always supported me, but my sister and brother haven’t, and my father even less*”.(A7)

Overall, they described completely satisfactory experiences, and did not express any pressure, obligation, or prohibition from their parents towards their sports practice. The only constraint reported by three athletes in total was the obligation from their families not to abandon their studies and balance the two activities. For example, Athlete A1 stated, “... obviously they didn’t allow their studies to take a back seat”. Athlete A5, when asked “If you had given priority to sports over studies?”, answered “I think there would have been a problem with my parents”.

Moreover, in the case of only one athlete, a manifestation of fear from their parents could be identified. However, this fear has nothing to do with sports practice or abandoning studies, but with the possibility of changing the athlete’s address and their inclusion in a sports residence in a different province: 

“*My mother was a bit scared, the motherly fear of your son leaving ... to a capital alone*”.(A6)

Family support was also indispensable in continuing their high-level sports career, as some of the interviewees stated that they were able to continue their sports career thanks to financial support from their families, as explained in the institutional-barriers category. Without this support, the athletes’ stay in Madrid would not have been possible.

#### 3.1.2. Coaches

All of the participants recognized the valuable work by coaches and the simultaneous determinant role they had played in their athletic development. In the dual-career process, they play the role of coaches and motivators to balance sports and studies or work. Some of the statements regarding this are as follows: 

“*Coach 1 said that studies come first and then sport*.”(A7)

“... *I would say that what coaches see is that with athletics you can’t earn enough money to live forever, so my coach... always told me that the education or training you do is more important than athletics*...”. (A3)

Perhaps a striking aspect of the relationship between coaches and athletes is the dimension that the relationship adopts. Three athletes described the relationship with their coaches as being like a parent/child relationship. Although the rest of the athletes did not mention the word “parent” or “mother” in their interviews, the way in which they referred to their different relationships could be interpreted in almost the same way:

“... *like my second family, as if they were my older brothers who take care of me, and my coach too*”.(A1)

#### 3.1.3. Educational Institutions

In general, all athletes mentioned support or great support from their institutions to balance training and competition with academic or work activities. This means being absent from the classrooms or not attending their job.

An athlete mentioned the figure of the “athletic tutor” at the university who helps in coordinating all activities and communication within the institution. However, it is striking that an athlete stated that an institute associated with the Madrid high-performance center was not the best place for an athlete, as the teachers did not care about the quality of education, which eventually becomes counterproductive for the athlete. This category is best understood by valuing the results of the educational-barriers category.

#### 3.1.4. Institutional

In general, all interviewed athletes agreed that there is limited support to develop high-level sports careers. Almost everyone had relied on some scholarship or financial support from the governing bodies of sport, such as the Higher Sports Council, the Royal Spanish Athletics Federation, and sports clubs, and the help of the different autonomous communities of Spain, but this is not continuous and fixed.

In this case, the experiences mentioned by the athletes demonstrate the low financial support they have:

“*Once I was given a scholarship, but it was suspended; it has been a year or so held back, it was the Olympic scholarship, it was 1000 euros one-time only and that’s it*...”.(A1)

“... *I was a scholarship recipient for one year when I was performing well, but the next year I didn’t perform well and they took away the scholarship, there were years that helped me and others that I was doing worse*...”. (A2)

### 3.2. Barriers

#### 3.2.1. Educational System

The interviewed athletes expressed that pursuing a dual career that does not neglect their studies while focusing on high-level sports is complex, and the educational system itself poses difficulties to this. A summary of the entire content of the category was expressed by Athlete A4: “...the demand is the same as for someone who does nothing else, and keeping up with people who have all day to do things...”

In this category, multiple obstacles were reported by the interviewed athletes, which demonstrates the difficulties that athletes face on a daily basis. Some examples are:

“...*yes, some of them understood me, but others always put up the obstacle of: “You have an advantage over your classmates*...”. (A6)

“...*the university also doesn’t provide many facilities for championships and that*...”(A5)

“... “*Couldn’t you do it at another time?”... or maybe, ...some competition exits, these trips, they looked at me with a bad face*”. (A1)

“...*I don’t know if it’s in all careers, but in my career, in the first year they force you to enroll in the entire course, and they don’t guide you very well on what you’re going to find. I mean, a whole course for someone who does sports -at least in my career, in other careers I don’t know- is too much, it’s too much of a load if you want to train well and if you want to have time to study and train*”. (A4)

“...*a teacher told me I had to choose between sports or studies, that the two things couldn’t be combined*”.(A6)

#### 3.2.2. Institutional Support

This section explains the institutional-support category and the consequences of the limited support in this category. In this situation, many athletes can abandon their sport to obtain a job that generates a stable income.

Uncertainty is the most recurrent factor for almost all the interviewed athletes, as scholarships and support depend directly on their athletic performance. 

A1: “forget it. You’re either an Olympic athlete, a world champion, an international medalist, or you don’t exist, and if you’re a thrower, even less so.”

In this category, all athletes reported different situations that did not allow for them to develop a successful athletic career without setbacks. This includes problems with money, accommodation, and continuity due to losing residency positions.

“...*In fact, I was there for one year; they usually give two, I mean, if you do badly one year they give you another chance, but I was there for one year and then they took away the scholarship because I didn’t have any results that year*”. (A4)

*Why do you lose that scholarship? “Because there are no big athletic achievements, to the demand they ask for*”. (A1)

“*Well, yes, I think so, to further develop this sport in particular, more support would be needed, or to structure it in a different way*...”. (A2)

### 3.3. Time Management

#### 3.3.1. Difficulty in Balancing Studies and Training

For all the interviewees, this was the most complex part. They all mentioned how difficult it was to meet all their daily commitments, such as attending classes, making deliveries, training, and resting.

“*Well right now I find it hard to sleep a lot, to fall asleep and have a deep sleep; ... then I often can’t disconnect, that’s a horrible thing and I think about things all day that are still left, ... in the end you arrive tired at training, you can’t train to the fullest, you don’t train well; then the things at uni also aren’t one hundred percent because you’re at half power*...”. (A3)

This category perfectly complements the educational-barriers category, as many of the expressed difficulties for perfectly balancing a dual career concern the academic workload and the demands of the studies, as expressed by the interviewees:

“... *I study veterinary—it has a lot of practical requirements and theoretical classes—you can be at the faculty from 9 in the morning until 7 in the evening with classes and practices, it cost me quite a bit, especially First which has a lot of workloads, it cost me and I have a lot left*”. (A4)

“*Organizing, especially, knowing at all times what you had to dedicate yourself to, you can’t be “I train, but I don’t study”, you have to have it clear*”. (A5)

“... *the higher degree is done in the morning and then in the afternoon I train normally and regularly, and on weekends if I can I work*”. (A7)

University students mentioned that most of the professors were aware of their sporting career, but they did not find unanimous support from them. One of the interviewees even mentioned that developing a dual career, referring to studies and training, was not a good approach for reaching the elite of the sport, and that they would discard their studies. 

“*I think that, if you strictly want a sports career to dedicate yourself professionally to sport, no, because in the end it prevents you from training, it prevents training sessions, it prevents resting well, if someone wants to make a strong sports career it’s not good*.” (A5)

#### 3.3.2. Management of the Pressures of the Dual Career

This category is associated with the difficulty in balancing studies and training.

One of the first issues that could arise in this category is the frustration of not being called up for a championship or not achieving the expected result, since it is the moment when the dual career of athletes is questioned. 

“... *then, yes, there was frustration, in fact, I considered quitting athletics, but this was at 24–25, already working after finishing the career and everything because I saw that I wasn’t rendering and the results didn’t come out, a lot of frustration*”. (A2)

Another important point is the processes that occur when changing categories and the additional pressure that comes with it. The athletes indicated that the level of demand is much higher, and the requirements to be called up by the national team are different and are not met. An example could be the pressure to perform athletically in order to renew economic support.

Several athletes agreed on the difficulty in adapting during the first year of changing their lifestyle and transitioning from a competitive to an elite sport or from the junior to the absolute category.

“*It was hard for me to adapt because actually, in the first year, I didn’t have good results.” ... “For me, it was difficult, I didn’t adapt. I didn’t adapt in the first year because I missed my parents, and then it was too big for me*...”. (A4)

“*Suddenly, being away from home, away from your friends, and in a town that is so familiar, the demands of the CAR. To be honest, I feel that the first year was too big for me, and the high performance, and the athletic demands*...”. (A1)

In this category, the topic of university and specifically the difficulty of combining a dual career with dependence on others arose again. This dependence refers to the need to add another item to the issue, the time of teachers and of organizing with classmates, as some do not understand the difficult task of balancing sports and studies.

“... *Right now I’m making that mistake again, that I want to do everything like everyone else, the whole course in one year, I think that with athletics training 6 times a week is a bit difficult and then I end up very stressed, and this year I saw that I already had quite a few [injuries], I’m sick again now and that comes from stress because I’m overloading myself*”. (A3)

### 3.4. Resources

Lastly, some resources emerged from the interviews that the athletes recognized as having helped them in balancing their athletic and academic lives. We split them into two categories, psychological and additional resources.

#### 3.4.1. Psychological Resources

There are numerous psychological factors that the athletes identified as a resource due to their positive impact in promoting good adaptation to the development of both athletic and academic careers. The most notable ones were motivation, ambition, failure management, anxiety management in competition, stress management from balancing both aspects of life, and confidence.

“... *Taking everything a little more calmly. I can’t do everything like the other students ... with athletics training 6 times a week, it’s a bit difficult and I end up very stressed. Right now I’m sick again and that’s because of stress because I’m overloading myself*”. (A3)

All athletes worked with a sports psychologist who helped them in managing the compatibility of their athletic and academic careers.

“... *I did learn a lot of techniques last year with the club psychologist. For example, the relaxation techniques helped me a lot when I got back from training, took a shower, had dinner, and was overwhelmed and tired from training. I would apply the techniques to get myself to study*”. (A6)

#### 3.4.2. Additional Resources

One of the most frequently cited resources by athletes to balance their athletic and academic careers is developing an academic career that helped them in relieving tensions related to athletic performance. In that way, having academic goals helps in not being constantly focused on the athletic aspect and ultimately, as A2 said, in achieving a “personal balance” that can help in athletic life. A4 commented on this:

“... *I think we need to have escape routes. Not only your life being athletics, having other routes for when it’s not going well, you can also focus on other things and your mind rests a little from this*.” (A4)

In addition, some participants said that, if you study for a university degree in a subject related to sports, this new knowledge may aid in better understanding one’s athletic self and the circumstances one is going through:

“*With the physiotherapy degree, I’m learning a lot. In the end, it makes you know many things, knows what’s right and what’s wrong, how you should recover or if you have an injury*”. (A5)

The athletes also identified that surrounding themselves with other athletes who had completed a dual career could serve as motivation and an example to follow. They also commented that playing sports gave them resources to persevere in other areas of life, such as academia, and that pursuing an academic career gave them an outlet for the possibility of their athletic career ending for any reason (e.g., lack of performance, injuries, pandemic).

## 4. Discussion

The purpose of this study was to examine the resources and barriers to combining sports and the academic life in the career of elite Spanish track-and-field athletes. The results support the need to frame the existence of barriers and resources for combining sport and studies within a holistic vision that takes into account the interaction of different spheres of life [8]. However, specific characteristics within track-and-field career development must be discussed.

The main barriers are related, on the one hand, to the difficulties posed by the educational institutions and the Spanish educational system structure in carrying out parallel academic and athletic careers and, on the other hand, to the elite athletes’ own time management. In the study of dual careers (DCs), the importance of the educational system and athletic/educational institutions in facilitating or hindering athletes’ ability to carry out a dual career was identified [21]. In this case, unlike other sports, the high-level athletic demand in track-and-field sport comes a bit later in terms of age, so all athletes in this study had finished or were finishing their secondary and high-school studies. However, the rigidity of the university curricula in Spain (e.g., class changes, exams, difficulty changing universities, and mandatory attendance) hinders reconciling academic and athletic life [19]. In this sense, policies that take into account the characteristics of elite athletes would facilitate them in not having to abandon their academic development due to its incompatibility with their athletic life. It would also be interesting to create adapted university models to this particular population.

One of the most important barriers is time management. According to the participants, the time needed to have a high-level athletic life and a university academic career are often in conflict, hence the importance of finding an optimal balance. Stambulova et al. [33] proposed two concepts related to the optimal balance of winning in both the short and long runs. In this case, to win in the short run, the athlete must prioritize on the basis of their life situation. For example, prioritizing university exams or sport in times of important competitions. Winning in the short run prepares the athletes to become winners in the long run (i.e., being ready for athletic retirement upon graduation or later).

The participants also identified a series of resources that helped them in facing the barriers and challenges of pursuing a dual career. First and foremost, support from the social environment. In that sense, the identified support came from families, coaches, the educational/work system, and institutions. Existing approaches in talent development research, such as the holistic ecological approach (HEA) [22,34], propose exploring the athletes’ environments to more reliably approach their realities. The athletic talent development environment (ATDE) proposes a model focused on the environment through a structure of two levels (micro and macro) and two domains (athletic and nonathletic). In the Spanish context, studies such as Mejias et al. [21] helped in identifying the support environments for the dual career of elite Spanish track-and-field athletes in public sports centers and private clubs. In this sense, the results of this study support this approach by considering the need for athlete support from a microathletic level (coaches), micro nonathletic level (families), macroathletic level (athletic associations), and macro nonathletic level (government and other institutions). Moreover, the need for these supports to develop dual careers was identified in other sporting contexts, such as football players [35,36], Olympic athletes [37], winter sports [38], and professional handball players [39].

In addition, the participants identified numerous psychological resources that helped them in coping with their sporting and academic careers. A psychological resource that emerged from the analysis of all interviews was motivation. Aunola et al. [40] described three motivational patterns in teenage athletes who were pursuing a dual career on the basis of their level of motivation for both sports and academia. Athletes belonging to the group, referred to as dual-career-motivated (high value of both sport and school), aspired to both obtain a university degree and be professional athletes. In a sample of young elite soccer players, Chamorro et al. [17] identified that those who valued both academic and athletic success had a more adaptive motivational pattern towards their sport than that of those who only felt successful with athletic achievements. So, being motivated towards both academia and sports could be a resource for dual-career athletes. Psychological variables also emerged that, when well-managed, become resources, such as ambition, failure management, anxiety management before the competition, confidence, and stress management related to balancing both spheres of life. Other studies, such as Brown et al. [16], suggested that self-discipline could be a good skill to cope with a dual career. Working with a sports psychologist on these variables, as indicated by the participants in this study, could be the key to transforming these psychological factors into resources and not barriers that impede the dual career.

Furthermore, the results show that pursuing a dual career can be an additional resource for development in both spheres of life. Although the HAC model proposes that different career development levels influence each other, there is little research on how academic development influences the athlete’s sporting sphere [8,21]. Participants in this study said that developing an academic career could help in relieving tensions related to competition and achieving a positive personal balance to face sporting challenges. In addition, pursuing university degrees related to sports can increase athletes’ knowledge about sporting factors that affect the development of their own athletic careers. On the other hand, as the scientific literature shows in studies on dual careers [8], pursuing academic and sporting careers in parallel can help in better managing a possible failure in the sporting career due to performance or health factors.

Therefore, it seems desirable to encourage elite Spanish track-and-field athletes to pursue a dual career. In this regard, the results of this study allow for us to establish a series of practical applications. First, we should encourage academic and sports institutions to remove the barriers that athletes mentioned in this study (academic flexibility, flexible schedules, and academic curricula). In this sense, the athlete’s environment (staff, sport psychologists, teachers, etc.) should work on time management skills to maximize the athlete’s ability to face the challenges of combining sports and studying. Another practical aspect would be to raise awareness among the athlete’s immediate environment (coaches, families, etc.) of the benefits of pursuing a dual career. In this regard, special care should be taken with coaches, as studies such as [41] showed that the coaches of individual sports are less supportive of the idea of pursuing a dual career. Lastly, psychological work should be incorporated into sports specialization, focusing on motivational, personality, and social variables that impact both the athlete’s sports life and other areas of life such as academia. Lastly, this study has some limitations. On the one hand, the sample was small. According to [42] when researching elite athletes, we observed a lack of an operational definition regarding the population to which the “elite” label refers. The elite athlete population is not numerous. Thus, the smaller the sample and the higher the competitive level of the athletes are, the greater the ecological validity and the clearer the definition of “elite” are. However, this entails assuming certain issues regarding the reliability and generalizability of the data obtained for the reference population. On the other hand, the sample may have been too specific in terms of geographical, cultural, and sporting contexts. Nevertheless, precisely because of the differences that exist in these aspects among the population of elite athletes, it would also be interesting to explore and gather information from small groups of participants, such as the one that comprised this sample.

## 5. Conclusions

This study showed that elite Spanish track-and-field athletes cope with barriers and resources when it comes to tackling the challenge of developing a dual career. Like in other sports, fundamental barriers are the rigidity of the Spanish educational system to allow for athletes to combine studies and sports, and the limited support from political and educational institutions. Moreover, the athletes indicated the difficulties of pursuing a dual career in terms of organizing dedicated time for both spheres of life, and the pressure that accompanies this combination.

On the other hand, in order to overcome this barrier, athletes are resourceful if social support is provided at both the micro (coaches, families, etc.) and macro (political and educational institutions) levels. Pursuing an academic career could help in alleviating inherent tensions to athletic life and finding a personal balance.

## Figures and Tables

**Table 1 ijerph-20-05153-t001:** Demographic details of participants.

Athlete	Age	Studies	Event
A1	21	Psychology	Hammer throw
A2	32	Marketing	Long jump
A3	25	Industrial engineering	Long jump
A4	28	Veterinary science	Long jump
A5	22	Industrial engineering	Sprint
A6	18	High School	Hammer throw
A7	20	High school	Long jump

**Table 2 ijerph-20-05153-t002:** Results of the study.

Dimension	Categories	Dimensional Description
Social support	Family	The responses included both support received by these groups, and the type of personal and formal relationships encountered in developing an optimal dual career.
	Coach	
	Educational/work	
	Institutional	
Barriers	Educative	Difficulties encountered by the athletes in the different groups in developing a dual career without prejudice to one of the two components.
	Institutional	
Time management	Managing dual-career pressures	The area encompasses the different ways of approaching the dual role of athlete/student or athlete/worker, and the conditioning of performing two equally important activities at the same time.
	Difficulty in balancing studies and training	
Resources	Psychological resources	This includes the mechanisms of facing the different situations of stress or pressure due to the workload of a dual career.
	Additional resources	

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
