# Peer review of "A Holistic Vision of the Academic and Sports Development of Elite Spanish Track and Field Athletes"

_ijerph, 2023, doi:10.3390/ijerph20065153_

Round 1

Reviewer 1 Report

Interesting idea for a study, considering that the usual focus for a sports/academics balance is U.S.-based athletes. And even those studies are rare.

The results section leans heavily on direct quotes without the authors contextualizing those quotes. However, a comprehensive discussion section can serve as the place for such context.

In addition, the transitions between the quotes often seem hastily written. For instance, at the top of Page 8 is this sentence fragment: “The pressures to perform athletically in order to achieve the renewal of the aforementioned economic support, etc.” The Results section needs more polish, as it is the heart of the manuscript.  

It’s a good idea for research, and it’s valuable to speak with athletes about their need to juggle multiple commitments outside of sport, and the support systems which help with that. However, with only seven athletes interviewed, it doesn’t quite feel “Big” enough for publication as a full manuscript. It’s a case Study in its current form. Improvements I would suggest, would include adding more subjects and providing a more comprehensive context within the Results section.  

Reviewer 2 Report

This paper studies a holistic vision of academics and sport development of elite athletes in track and field. This is an interesting topic due to the complexity of elite athletes’ situation, and the difficulties that can turn up during their lifespan.

In order to clarify some aspects, the following suggestions are included below:

Abstract

It is well written and organized.

Keywords

I see appropriate including keywords like for example “high performance” or some other keywords to fit better to the content.

Introduction

There is a good analysis about the situation in the introduction, but the manuscript could be improved adding a more extended information about barriers and resources.

A paragraph and significant references are needed in order to justify two of the main construct of this research,…barriers and resources  (please justify with references).

More examples about research developed in another sports, individual and team sports should be included in the introduction, and then discussed at the discussion section.

Methods

-          In the data analysis section, a more detailed information about IPA should be included, specifying the 6 steps of this methodology. Moreover, references using such methology should be included, apart from Smith et al 2017.

Discussion and

-          The new references included at the introduction section should be also mentioned and discussed in this discussion section.

-          Limitations of this research and future possible investigations should be included in order to improve the discussion section and the quality of this manuscript.

Conclusions

References:

Line 457: parents “y” coaches?

Vanden Auweele, Y.; De Martelaer, K.; Rzewnicki, R.; De Knop, P.; Wylleman, P. Parents y coaches: A help or harm? Affective outcomes for children in sport. In Ethics in youth sport; Y. Vanden Auweele.Ed; Lannoocampus: Leuven, Belgium, 2004.

The reviewer appreciates the interest of this contribution and believes that with suitable changes this paper could be published in Ijerph

Reviewer 3 Report

This paper is mainly written with clarity and the empirical evidence is used engagingly in section 3 – I was pleased to see some indication of the strength of opinion, even in a small sample. The set-up in section 1 is well argued and research design described adequately. Section 4 is the element of the manuscript that I think needs strengthening (see below).

Generally,

Is there something to say about the Spanish context? Although the data are gathered from Spanish athletes, is there anything transferable to other contexts? Are there particular circumstances in Spain that make this a unique situation?

Was there any verification of the transcripts and/or analysis from the participants?

The Discussion (section 4) needs to make more explicit the new contribution to theoretical knowledge and/or professional practice on the basis of the findings of the study.

 There are some specific points:

31           It would help for the international audiences to make clear that this is ‘association football’ (soccer).

38 & 85                 The DC does not seem to be consistent with the emphasis of the study of academic / sporting duality. Disambiguation would help.

48           Would a visual representation of the HAC help the readership to understand it?

64           Should this be ‘… athletic career development within the academic career development of athletes.’ ?

70           See, perhaps, the work by Harry Bowles (2018) on cricket in the UK: University Cricket and Emerging Adulthood: Days in the Dirt. (Palgrave Macmillan)

90           What does ‘high school’ mean in this context – especially for the 20 year old, A7?

Table 1                  A7 Long jump

95           Not usual to refer to a semi-structured interview as an instrument.

98-100   Suggest this might be better in section 1.

107         Block of what?

121         'withdraw' rather than ‘opt out’?

136         To what/whom does triangulation refer? Confirmation across the research team? If only those themes over which there was unanimous agreement were included, what about the others? Were there any?

163         The ‘fear or fear’ point is unclear (at least to me).

359         DC?

361         Absent references

363         Later than what/whom?

The paper also needs a ‘copy-edit’ for precision in the use of English – e.g.,

12, 15    use of ;

Section 3              Use of tense in the presentation of data. Usual not to be in the ‘present’.

Round 2

Reviewer 2 Report

Thanks for the improvements, the article is much better now.